# Rheology of Polyacrylonitrile/Lignin Blends in Ionic Liquids under Melt Spinning Conditions

**DOI:** 10.3390/molecules24142650

**Published:** 2019-07-22

**Authors:** Jinxue Jiang, Keerthi Srinivas, Alper Kiziltas, Andrew Geda, Birgitte K. Ahring

**Affiliations:** 1Bioproducts, Sciences and Engineering Laboratory, Washington State University, Tri-Cities, 2710, Crimson Way, Richland, WA 99354, USA; 2Director of Chemistry and Technology, Wood Protection Technologies, Eco-Building Products, 11568 Sorrento Valley Rd, San Diego, CA 92121, USA; 3Fermentation Scientist II, Lygos Inc, 1249 Eighth Street, Berkeley, CA 94710, USA; 4Research and Innovation Center, Ford Motor Company, Dearborn, MI 48124, USA; 5Hyundai-Kia America Technical Center, Inc., Superior Charter Township, MI 48198, USA; 6Biological Systems Engineering, L.J. Smith Hall, Washington State University, Pullman, WA 99164, USA; 7The Gene and Linda Voiland School of Chemical Engineering and Bioengineering, Washington State University, Pullman, WA 99163, USA

**Keywords:** biorefinery lignin, carbon fiber, dynamic rheology, ionic liquid, polyacrylonitrile

## Abstract

Lignin, while economically and environmentally beneficial, has had limited success in use in reinforcing carbon fibers due to harmful chemicals used in biomass pretreatment along with the limited physical interactions between lignin and polyacrylonitrile (PAN) during the spinning process. The focus of this study is to use lignin obtained from chemical-free oxidative biomass pretreatment (WEx) for blending with PAN at melt spinning conditions to produce carbon fiber precursors. In this study, the dynamic rheology of blending PAN with biorefinery lignin obtained from the WEx process is investigated with the addition of 1-butyl-3-methylimidazolium chloride as a plasticizer to address the current barriers of developing PAN/lignin carbon fiber precursors in the melt-spinning process. Lignin was esterified using butyric anhydride to reduce its hydrophilicity and to enhance its interactions with PAN. The studies indicate that butyration of the lignin (BL) increased non-Newtonian behavior and decreased thermo-reversibility of blends. The slope of the Han plot was found to be around 1.47 for PAN at 150 °C and decreased with increasing lignin concentrations as well as temperature. However, these blends were found to have higher elasticity and solution yield stress (47.6 Pa at 20%wt BL and 190 °C) when compared to pure PAN (5.8 Pa at 190 °C). The results from this study are significant for understanding lignin–PAN interactions during melt spinning for lower-cost carbon fibers.

## 1. Introduction

Significant improvements have been made in blending lignin into different polymers in order to reduce cost, increase biodegradability, or to produce specific properties of importance for the applications of the specific polymer. Lignin is the second most abundant renewable resource after cellulose in the terrestrial ecosystem and is typically available as a by-product from wood pulping and cellulosic biorefineries. Several polymer/lignin blends have been reviewed, related especially to their self-interactions, interactions with the polymer, and their effect on the properties of the final product [1,2]. These studies focused on the interdiffusion of the lignin into the polymer, its effect on the mutual solubility of the different phases, and the strength of the interfacial adhesion between components in heterogenous blends. This effect is particularly important in situations where the blending can affect both the spinning/casting of the polymer and the mechanical properties of the final polymeric product such as carbon fibers.

The unique properties of high strength and modulus, low density, excellent creep, and chemical resistance make carbon fiber an emerging material for manufacturing advanced composites which have application for the automotive, aerospace, windmill, and sporting goods industries [3]. However, a widespread application of carbon fibers is still limited by the high cost of petroleum-based polyacrylonitrile (PAN), which contributes to 51% of the total cost for carbon fiber productions [4]. Therefore, it is crucial to lower the cost of carbon fibers either by developing inexpensive alternate precursors or by in-mixing low-cost materials into PAN, which will bring down the overall cost while still keep the advantages of PAN as a precursor for carbon fibers.

Lignin is a promising candidate for producing carbon fibers because of its carbon-rich aromatic chemical structure and low cost. Extensive research efforts have been invested in manufacturing carbon fibers with 100% lignin (from kraft, sulfite, and organosolv processes) as the primary carbon source [3]. However, the production of precursor fibers for carbon fiber production based on lignin alone has major challenges. The complex heterogeneous structure of lignin with randomly cross-linked phenylpropane units in a 3D network makes it difficult to spin and spool into fibers without modification. It was also found that the lignin-based carbon fibers had a lower tensile strength and modulus and did not meet the performance targets (tensile strength of 1.72 GPa and modulus of 172 GPa) set by the US Department of Energy (DOE) for various applications [4,5].

Recently, it was proposed that blending (or in-mixing) of lignin with polymers such as polyacrylonitrile (PAN) can overcome some of these problems and also reduce the overall processing cost of carbon fibers. Several studies have been done using low-temperature spinning techniques such as solution spinning [6] and electrospinning [7,8] for producing PAN-lignin precursor fibers. Industrial-scale production of lignin/PAN carbon fiber by solution-based wet spinning technology has been explored by Zoltek and Weyerhaeuser together under a DOE contract [6]. They successfully obtained a carbon fiber with a tensile strength of 2.24 GPa (or 325 Ksi) and modulus of 216.72 GPa (or 31.5 Msi) by incorporating 25% lignin in the precursor fiber blend. However, these solution-based spinning technologies usually require low solid concentration in dope, and hence involves a large amount of organic solvent, which can lead to significant environmental issues and process costs. Also, several of these studies have shown that at solution spinning conditions, there is only a physical interaction between the polymer, and no chemical interaction with the lignin, resulting in problems with spinnability of the fibers [7], porous weaker structure [9], phase separation [10], and reduced draw-ability [11]. Liu et al. [12] have shown that PAN can be blended with softwood lignin using gel spinning, resulting in high-tensile strength carbon fibers. However, they found that a decrease in the activation energy of PAN occurred when lignin was added due to increased cyclization, cross-linking, and oxidation. The blending of lignin with PAN using melt spinning can also reduce the usage of large quantities of toxic solvents needed and result in higher production efficiency.

However, direct melt-spinning of lignin/PAN blends is challenging primarily due to the high melting point of around 300 °C for PAN, and PAN begins to undergo decomposition at these temperatures. Also, at temperatures around 220 °C, PAN undergoes a series of complex intra- and inter-molecular cyclization reactions primarily due to its C≡N groups preventing thermoplastic fabrication and resulting in a non-conformational rheology at melt spinning conditions, thereby requiring internal or external plasticization [13]. Internal plasticization has shown greater promise with copolymers such as methyl methacrylate [14], itaconic acid [15], ammonium acrylate [16], etc., of which, methyl methacrylate has been found to be the most promising internal plasticizer. However, these studies also indicated that while 85% PAN-15% methyl methacrylate showed best melt spinning performance to produce carbon fibers, the overall process efficiency was significantly reduced due to low carbon conversions [6].

External plasticization can also be used to decouple the nitrile–nitrile interactions resulting in a lowering of the polymer softening point below its decomposition temperature and making a pseudo-melt spinning process. Some of the commonly used plasticizers include water, dimethyl formamide, ethanol, and acetonitrile, or a mixture of them [17]. These studies produced a melt-spun fiber with water/acetonitrile as a plasticizer with a fiber strength of 4.472 GPa (or 650 Ksi) that fulfilled DOE requirements. However, a piece of special equipment with a pressure chamber installed is required to prevent evaporation of such solvents during the melt-spinning process. There has been a growing scope for application of ionic liquids in polymer production, and several studies have investigated rheology and spinning properties of PAN in several ionic liquids such as 1-butyl-3-methylimidazolium chloride [BMIM]Cl [18,19,20] and 1-butyl-3-methylimidazolium bromide [21]. These studies indicated that the ionic liquids increased the apparent viscosity of PAN, resulting in higher polymeric entanglements and shear thinning behavior at higher shear rates. The PAN-[BMIM]Cl system has been previously used in the production of carbon fibers through melt spinning, which resulted in a final strength of the carbon fibers which was almost twice as that produced through wet spinning [11]. These studies also showed that the low imperfect morphology and a cyclized structure formed by in situ chemical reactions during the melt spinning process contributes to the increased fiber strength and decreased residence time for down-step thermal stabilization. The greatest advantage of using an ionic liquid such as [BMIM]Cl is that it can also effectively dissolve lignin without need for any modification [22]. However, previous studies have indicated that the presence of hydroxyl groups in the lignin prevents it from optimal chemical modification with polymers [9], and hence, some sort of lignin modification such as esterification will be required for effective chemical interaction between PAN and lignin in [BMIM]Cl. Previous studies have shown that lignin esterification facilitates the molecular thermal mobility and reduces intermolecular interactions within the lignin structure [7]. In order to produce carbon fibers from lignin/PAN blends in [BMIM]Cl using melt spinning, it is very important to initially understand the thermal and rheological properties of the polymer blends.

However, most studies discussed above have, predominantly, used purified lignin streams for blending with polymers. These purified lignin streams are usually obtained after removal of cellulosic components present in the biomass, which usually requires toxic solvents or extreme reaction conditions, which can also result in thermal degradation of the lignin. Such separation and purification processes can affect the structure and properties of lignin, which can affect the quality of the final product. For example, previous research using the alkaline process for lignin production has shown cleavage of β–*O*–4-linkages in the proto-lignin structure, resulting in the onset of complex condensation processes that resulted in weaker interactions with polymers and eventually the lower tensile strength of the final product [1]. The use of a lignin stream directly from a biorefinery with minimal washing has not previously been attempted and holds significant advantages with respect to feedstock costs. The presence of other biomass components can also increase intermolecular interactions between the biorefinery lignin and PAN polymer. Hence, in this study, the rheology of PAN with different concentrations of biorefinery lignin in [BMIM]Cl has been studied to understand the effect of temperature on the interactions between the components during melt spinning to produce carbon fiber precursors. The incorporation of the biorefinery lignin into PAN had a significant impact on its rheology and melt strength by introducing cross-linking or polymer entanglements. The biorefinery lignin was characterized to provide sufficient information to help understand the effect of the different components on blending with PAN at melt spinning conditions.

## 2. Results

### 2.1. Lignin Characterization

Biorefinery lignin (Table 1) was found to contain around 30%wt cellulose (in the form of glucan) along with lignin. The structural characteristics of lignin after wet explosion pretreatment has been previously published [22,23], and hence, this section will focus on the specific structural properties of the biorefinery lignin substrate that was blended with PAN.

As can be seen from Table 1, the molecular weight of the biorefinery lignin is high and indicates the presence of cellulosic components in the lignin. This was further confirmed by solid-state NMR analysis of the butyrated biorefinery lignin stream, as shown in Figure 1. Prominent cellulosic components in ^13^C-NMR data shown in Figure 1 can be usually found at 82, 89, and 105 ppm, which refers to C_6_, C_4_, and C_1_ carbon atoms in carbohydrates, respectively. Similarly, the most common lignin-based linkages in the ^13^C-NMR data were usually present at 30–40 ppm and 56 ppm, which refers to alkyl CH & CH_2_ in lignin subunits and methoxy groups in lignin’s guaiacyl & syringyl units, respectively. As can be seen from Figure 1, there are ester or carboxylic linkages (169–195 ppm), which is significant since previous studies have shown that these linkages are primarily responsible for cross-linking during oxidative thermo-stabilization [23,24]. These studies have also indicated that during polymer devolatilization/extrusion process, chain scission occurs primarily at the methoxy groups and at the aliphatic side chains, resulting in degradation and condensation reactions [24]. Due to the higher amount of methoxylation in the biorefinery lignin obtained from wet explosion pretreatment [25,26], the butyrated biorefinery lignin stream (hereby, referred to as “BL” for convenience sake) used in this study can, hence, be a promising feedstock for blending with PAN to produce carbon fibers.

### 2.2. Effect of Temperature and Lignin on Viscosity of PAN

The apparent viscosity of PAN and PAN-BL blends in [BMIM]Cl was measured as a function of angular frequency at different temperatures and shown in Figure 2a–d. It can be seen from Figure 2a that the apparent viscosity of the PAN solution was predominantly linear as a function of angular frequency until around 10 rad/s, followed by a slight decrease with increase in angular frequency. This decrease in viscosity at a higher frequency can be attributed to the pseudo-plastic behavior of PAN. This behavior is especially prominent at higher temperatures (>190 °C).

In order to study the effect of BL in-mixing (or blending) on the melting temperature of PAN, differential scanning calorimetry (DSC) measurements were done as shown in Figure 3. It can be seen from Figure 3 that the melting temperature for PAN in [BMIM]Cl was around 116 °C. Previous studies have shown that an increase in [BMIM]Cl concentration to PAN resulted in a reduction of its glass transition temperature (T_g_) and melting temperature (T_m_) [13]. These studies, however, showed that significant impact of [BMIM]Cl on PAN was on its crystallization process due to inhibition of the cyclization reactions of C≡N bonds present in PAN by [BMIM]Cl, especially at temperatures greater than 200 °C. Hence, the temperatures used in this study were below 200 °C to minimize the effect of such cyclization reactions and to directly study the effect of BL on PAN melts. Figure 3 also showed that while BL did not have a significant impact on the melting temperature of the PAN-[BMIM]Cl system until around 20%wt, an increase in BL concentration to 30%wt did introduce a peak anomaly at around 85 °C which can be directly attributed to the BL peak (as shown in Figure 3). This BL peak at 30%wt blending with PAN can be indicative of a heterogeneous melt system that can reduce the efficiency of the melt spinning of the PAN-BL blends. However, an opposite effect was observed by Oroumei et al. [8] while studying the effect of lignin blending with PAN in dimethyl formamide solvent. These studies showed a perfect blending of PAN-lignin at all concentrations up to 90% wt. These studies did, however, also report that addition of lignin resulted in an increase in the T_m_ of PAN and also showed broader DSC peaks that are, generally, indicative of molecular-level interactions between PAN and lignin which occur during thermal treatments. Similar observations were found in our study, where the T_m_ was measured to be 116.3 °C and 130.5 °C for pure PAN and 10%wt BL in PAN, respectively. However, an increase in BL concentration to 20 and 30%wt in PAN reduced the T_m_ to 117.2 and 120.8 °C respectively, which was still higher than PAN fibers, but showed evidence of multiple peaks within the temperature range (heterogeneous system).

The dynamic viscosity at zero shear for the PAN and PAN-BL blends can be calculated from the complex viscosity using the Cox–Merz rule when angular frequency (ω) is equal to zero [27,28]. The effect of temperature on the zero-shear viscosity (η_0_) of the PAN/BL blends can be better understood using the Arrhenius solution activation energy model (Equation (1)) where ln (η_0_) is plotted as a function of temperature with the activation energy obtained from the slope of the curve (Figure 4). Studies have indicated that the Arrhenius relationship can be used to describe the polymer chain restrictive force to flow using the activation energy of the flow [29].
(1)η0=AeEaRT
where *A* is the Arrhenius constant, *E_a_* is the solution activation energy, *R* is the universal gas constant, and *T* is the absolute temperature (K).

The linear relation between logarithmic exponent of zero-shear viscosity and the inverse of absolute temperature at which it is measured is an indication of a decreasing viscosity of the blends with an increase in temperature and decrease in lignin concentration. Previous studies on PAN-lignin blends in organic solvents such as dimethyl formamide showed a decrease in blend viscosity with an increase in lignin concentration, both before [28] and after butyration [7]. The difference in behavior of the lignin can be attributed to the difference in the properties of the solvents where DMF is more volatile and non-polar in nature when compared to a comparatively polar ionic liquid such as [BMIM]Cl. From Figure 4, the solution activation energy for PAN was estimated to be 51.7 kJ/mol, while that for PAN/BL blends was found to be 46.1 kJ/mol, 43.3 kJ/mol, and 32.9 kJ/mol at 10, 20, and 30%wt BL, respectively. The decreasing trend in the solution activation energy of the PAN/BL blends with an increase in BL loading can be attributed to heterogeneity in the system with an increase in BL concentration and is later discussed in detail in this manuscript.

### 2.3. Effect of Temperature and Lignin on Dynamic Rheology

The storage and loss modulus plots as a function of temperature during both the heating and cooling scans are shown in Figure 5. As expected of thermo-reversible polymers [27], the loss modulus (G′′) was greater than the storage modulus (G’) in the entire temperature range showing a liquid-like behavior. There are a few studies on the dynamic rheology of PAN in ionic liquids such as [BMIM]Cl [18,20] or [BMIM]Br [21], or other plasticizers such as dimethyl sulfoxide (DMSO)-water mixtures [30] and dimethyl formamide (DMF) [28] at much lower temperatures (for solution spinning & electrospinning) when compared to this study. However, the observed trend of G′′ > G′ as a function of temperature for PAN in [BMIM]Cl was previously seen at temperatures up to 120 °C [20], which further supports the rheological data discussed in our study. The variation in the G′′ and G’ as a function of the angular frequency followed a similar trend as that of the complex viscosity discussed above and was found to be significantly affected by the temperature and BL composition (Figure A1). Hence, the variation in G′ and G′′ as a function of angular frequencies at different temperatures and BL compositions are discussed in Appendix A of the manuscript.

An increase in BL in-mixing with PAN showed a deviation from the thermo-reversibility behavior of the PAN, i.e., deviation of G′ and G’’ for the heating/cooling curves, and this trend was also previously observed when DMF was used as solvent and at much lower temperatures up to 60 °C [27]. Hence, this non-conformational behavior of lignin as a blending agent with PAN is not specifically temperature-dependent but is significantly dependent on the concentration of lignin. It can be seen from Figure 5b–d that at 10%wt BL, there was only a slight deviation in thermo-reversibility of the polymer blend but with an increase to 20 & 30%wt BL, both G’ and G′′ did not exhibit thermo-reversibility. This trend was also previously observed by Liu et al. [27] and was attributed to a strong dynamic asymmetry of PAN and BL in [BMIM]Cl. It can also be seen from Figure 5 that, while not seen for PAN, there was a crossover point between G’ and G′′ around 195 °C, 192 °C, and 188 °C for PAN with 10, 20, and 30%wt BL blending, respectively. This crossover point has been, traditionally, considered to occur due to thermo-rheological interactions between the PAN and BL resulting in a physical gel network and the temperature at which crossover happens is considered as “gel point” [27] and this gel point temperature for PAN in an organic solvent system such as DMSO-water was found to be between 60–70 °C [29]. Hence, it can be seen that an increase in the lignin concentration induces polymer-gel formation at higher temperatures, which can mean an increased entanglement of the polymer chains, reducing mobility of the polymer blends during heating and the cooling process especially [30]. It is thus ideal to melt spin PAN-BL blends at temperatures below their respective gel points and melt spinning at the same temperature that was optimized for PAN in [BMIM]Cl can result in reduced performance of the PAN-BL fibers. This observation is very significant for the optimization of the melt spinning conditions when blending BL with PAN.

The log-log G’ vs. G′′ plot, also known as Han plot, has been widely applied to understand and interpret the homogeneity and polydispersity of polymer solutions [31,32,33]. The Han plots for the PAN and PAN-BL blends in [BMIM]Cl are shown in Figure 6a–d, and slopes of the Han plots at different temperatures and BL concentrations are shown in Table 2. It can be seen from Figure 6 that, at every blend compositions, the log G’ vs. G’’ plot is higher than that for pure PAN, but the trends are very similar between 10 and 20% PAN-BL blends and significantly increased for the PAN-30%BL blends. This sudden increase with increase in BL concentration to 30%wt, while it can be attributed to the amplification in the differences in polydispersity of PAN and BL, would probably be more related to the heterogenous melt system at that BL concentrations discussed earlier. While polymeric in nature, butyrated lignin (BL) itself does not supply any specific polymeric properties to the PAN homopolymer, but the significance of the study is that the results indicate that BL does not simply act as a particulate filler to the PAN, instead forming a homogeneous melt system, at least until 20%wt BL concentrations.

In order to further understand this phenomenon, the yield stress [34] was calculated using modified Casson model (Equation (2)), and its plot as a function of temperature is shown in Figure 7.
(2)$$\sqrt{G^{\prime \prime }}=\sqrt{\sigma_{0}}+\sqrt{k\omega }$$
where G′′, σ_0_, k, and ω refer to the solution loss modulus, yield stress, viscosity coefficient, and angular frequency, respectively. The yield stress is an indication of the structural breakdown of the viscoelastic liquid. It has been previously observed that the yield stress usually decreases with an increase in temperature and decrease in lignin concentration for PAN-lignin blends at temperatures up to 60 °C [28]. However, as can be seen from Figure 7, the yield stress at each temperature significantly increased with an increase in lignin concentration, indicating the ability of PAN-BL blends to withstand higher break-up stress when compared to PAN. It can also be seen from Figure 7 that while σ_0_ for PAN decreased with an increase in temperature, the PAN-BL up to 20%wt BL showed independence of σ_0_ as a function of temperature, but this independence failed significantly at 30%wt BL. Hence, it can be seen that at melt spinning temperatures, there are significant polymeric and intermolecular interactions between PAN and BL, resulting in an increased viscoelastic performance during melt spinning, especially below the gel point.

## 3. Discussion

The goal of this study was to understand the effect of melt spinning conditions on the PAN-lignin (butyrated) blends for application in carbon fiber production. The glass transition temperature (T_g_) of BL was found to be around 103.2 °C (149.2 °C before butyration) which was found to be slightly higher than that reported in the literature. This is to be expected since the T_g_ would be significantly dependent on the lignin structure, type of treatment/purification, and even moisture content. Kadla and Kubo [35] reported T_g_ of hardwood Kraft lignin to be around 108 °C. Other studies [6] indicated a lower T_g_ of 71.5 °C for organosolv lignin after butyration. Most of these lignins were found to have a lower molecular weight than that used in this study. The T_g_ of softwood kraft lignin fibers with a molecular weight between 3.3 and 6.4 kDa was found to be around 140 °C (without esterification), which is very similar to T_g_ of 149.2 °C for biorefinery lignin measured in this study [36]. The typical T_g_ for PAN is around 85 °C while its melting temperature can be as high as 300 °C [7]. The higher T_g_ of BL can have a significant impact on the rheology properties of the PAN.

It can be seen from Figure 2a that the complex viscosity of the PAN/BL blends in [BMIM]Cl decreased with an increase in temperature, which can be indicative of an increased spinning ability and elasticity of the polymer. Similar trends were also observed by Liu et al. [20] even though they studied the rheology of PAN in [BMIM]Cl at lower temperatures (<120 °C). These studies indicated that development of elasticity in a polymer solution with an increase in temperature could produce phenomena such as die swell of “puff up” of fibers after spinning and is also closely related to molecular interaction in molded objects. It can be seen from Figure 2b–d that an increase in BL concentration resulted in an increase in the complex viscosity when compared to pure PAN. This increase in viscosity could be related to the effect of BL blending on the melting temperature of PAN, which results in a reduction of its elastic properties. However, an increase in BL blending (say, at 30% wt, Figure 2d) almost showed a linear decrease in the apparent viscosity of the PAN-BL blend as a function of angular frequency. As indicated previously, this trend can be attributed to the pseudo-plastic behavior of PAN at concentrations of BL greater than 20% wt.

The activation energy of PAN in [BMIM]Cl (from Figure 4) is closer to that previously reported in the literature [18,20]. However, it can be seen from Figure 4 that there is a slight deviation from the Arrhenius relationship with the addition of BL to PAN. This deviation is not consistent but seems to be prominent above 170 °C. Liu et al. [20] indicated that high molecular weight polymer solutions have a greater tendency to form entanglements that can result in a deviation from Newtonian behavior, and the trends for PAN-BL blends above 170 °C seem to follow this deviation. This deviation can also be related to the increased concentration of PAN in [BMIM]Cl as observed by Liu et al. [20] where concentrations above 22%wt PAN in [BMIM]Cl were found to show similar deviations at high shear rates. In accordance with this argument, there is a slight but non-negligible deviation that can be observed for PAN in Figure 4. This deviation is amplified with an increase in BL concentration in lignin; this can be attributed to the initiation of a previously discussed heterogeneous system for the blends especially above 20%wt BL concentration.

For further understanding of the chemical and physical properties of these blends, the loss tangent (tan δ) for PAN and PAN-BL blends was plotted as a function of temperature (Figure 5). Unlike the trend observed by Liu et al. [27], an increase in BL concentration resulted in a decrease in tan δ for both heating and cooling scans. However, the difference in the observations can be attributed to the higher temperatures and ionic liquid used in our study when compared to Liu et al. [27]. It can be seen from Figure 5a that the tan δ for PAN was higher during cooling than heating at temperatures above 185 °C. This is an indication that at temperatures below 185 °C, the loss modulus decreased while cooling when compared to heating as a function of the storage modulus. A similar trend was observed at around 190 °C when 10%wt BL was added to PAN but did not specifically show up at higher concentrations of BL. Previous studies [18] have indicated that a super-cooling of the liquid in concentrated PAN/[BMIM]Cl solutions occurred at temperatures below 70 °C due to a decrease in interaction between the C≡N and [BMIM]^+^ cation during cooling. However, there are no studies that have observed or discussed these trends when lignin was added to the PAN-[BMIM]Cl solutions. While several conclusions can be made from this trend on the physical and chemical interactions of PAN and BL in [BMIM]Cl, further studies are required to completely understand the nature of this blend system and its effect on the final strength of the carbon fibers.

Traditionally, a slope of 2 in the Han plot is an indication of a homogeneous, monodisperse polymer solution or melt, and that less than 2 is an indication of a heterogeneous melt system [37]. It is also known that the Han plot is not dependent on the melt temperature and the M_w_ of the monodispersed polymers [13,38]. Previous studies done on PAN in DMSO-water mixtures [30] showed a slope of 1, which was indicative of a heterogeneous melt system and was independent of water content. Similar studies done on PAN-lignin blends in DMF showed a slope of 1.57, and this value was also independent of temperature or lignin concentration [8]. Noting that both these studies were done at lower temperatures when compared to the current study, it can be seen from Figure 6 and Table 2 that the slopes of the Han plots were significantly dependent on both BL concentration and temperature. The slope was found to be around 1.47 for PAN at 150 °C, which is closer to that observed previously [27] and decreased with an increase in BL concentration and an increase in temperature. The small difference in the slope for PAN in [BMIM]Cl when compared to that in DMF [27], with the difference in measured temperatures considered, can indicate a minor effect of the solvent on the homogeneity of the PAN-polymer system. However, the interesting trend is the decrease in the slope with an increase in BL concentration, which has previously indicated an increase in elasticity and melt strength [36,39] due to an increase in cross-linking, branching, and even intermolecular interactions. However, there are not many studies that have found a slope less than 1, and since this happens at higher temperatures (≥ gel point) and higher BL concentrations in PAN, it can be related to the polymer entanglements (mostly only in case of polymer mixtures and not homopolymers) as previously discussed. Other studies have also attributed increased spreads in the Han plots with the polydispersity of the components in the polymer blends [39,40]. These studies have indicated that if the monodisperse components in the polymer blends have dissimilar chemical structures, there is a higher ability for the log G’ vs. G’’ plot to be higher for certain blend compositions when compared to the individual components.

## 4. Materials and Methods

### 4.1. Materials

Polyacrylonitrile-*co*-methacrylic acid (PAN-*co*-MAA; 5.8wt% MAA, Mw = 85,000 g/mol) was obtained from Goodfellow, Co. (Coraopolis, PA, USA). The ionic liquids 1-butyl-3-methylimidazolium chloride [BMIM]Cl, 1-methylimidazole (1-MI), and butyric anhydride were obtained from Sigma-Aldrich (St. Louis, MO, USA). Biorefinery lignin was provided by CleanVantage LLC. (Richland, WA, USA) and was obtained after wet explosion pretreatment and enzymatic hydrolysis of poplar (*Populus*) sanders dust. The biomass was converted into sugars, resulting in a by-product stream, the biorefinery lignin, which was washed in water and ethanol.

### 4.2. Chemical Modification of Lignin

Biorefinery lignin was esterified using a modified butyration method [7]. In this process, 1 mL of *N*-methylimidazole as catalyst was added to 10 g of lignin along with 75 mL of butyric anhydride and reacted at 65 °C in a round bottom flask. The mixture was stirred 24 h under reflux, and Deionized (DI) water was added to quench the reaction. After natural cooling to room temperature, the precipitated lignin was vacuum-filtrated and washed three to five times with 500 mL DI water to remove catalyst and free acid. The modified lignin was then freeze-dried for at least 3 days. The dried lignin powder was stored in a desiccator before use.

### 4.3. Lignin Characterization

The compositional analysis (sugars and lignin) of the biorefinery lignin was done using laboratory analytical procedures developed by National Renewable Energy Laboratory (Golden, CO, USA) [41]. Composition of carbon, nitrogen, and hydrogen in the biorefinery lignin was measured using CHN628 (Leco Corporation, St. Joseph, MI, USA). The nuclear magnetic resonance (NMR) analysis of the biorefinery lignin after butyration was done using DMSO-d6 as solvent (Cambridge Isotope Labs, Woburn, MA, USA). The ^13^C cross polarization-magic angle spinning (CP/MAS) NMR analysis has been previously used [24]. The carbon spectra were acquired with a sweep width of 34,722 Hz using an acquisition time of 0.4 s and a relaxation delay of 1.6 s. A 30° pulse was used, and broadband 1H decoupling was used only during the acquisition time. A total of 30,000 scans were recorded for each spectrum, and the free induction decay (FID) was apodized with 12 Hz of exponential line broadening prior to zero filling to 65K points and Fourier transformation. An acquisition time of 2.0 s, a relaxation delay of 3.5 s, and a 45° pulse were used to collect 64 scans for each spectrum. The FID was apodized with 0.5 Hz of exponential line broadening prior to zero filling to 65K points and Fourier Transformation. Spectral widths of 6250 and 24,125 Hz were used for the ^1^H and ^13^C dimensions, respectively. An acquisition time of 0.199 s was used to directly observe dimension, and an acquisition time of 0.0066 s was used for the indirect dimension, and 48 scans were taken per increment. A one-bond ^1^H–^13^C J coupling of 150 Hz was used, and a total of 2 × 160 increments in t1 were acquired using the gradient echo-anti echo selection technique for pure phase line-shape in F1. The FIDs were zero-filled once to 2048 points in t2, apodized with a Gaussian function prior to Fourier transformation. Data in t1 were extended by a factor of 2 with linear prediction followed by zero filling to 2K points, apodizing with a Gaussian function and Fourier transformation.

The butyrated biorefinery lignin samples were dissolved in DMSO and molecular weight analysis was done using size exclusion chromatography (SEC) using a Shimadzu HPLC (LC-10AD, Kyoto, Japan) equipped with a Phenogel column (GP/4446, 300 mm × 7.8 mm; Phenomenox, Torrance, CA, USA), refractive index detector (RID 10A, Shimadzu, Japan), and an SCL-10A data station. The SEC analysis was done using tetrahydrofuran as a mobile phase flowing at 1 mL/min at 30 °C. The molecular weight of lignin was determined using polystyrene calibration standards.

### 4.4. Preparation of Melt-Spinning Blends

The preparation of PAN melt using ionic liquid [BMIM]Cl as external plasticizer has been previously described [11]. The weight ratio of PAN to [BMIM]Cl was set as 50:50 for all the different blends. A ball mill was used to facilitate homogeneous mixing to produce PAN-lignin blends at lignin concentrations varying between 10 and 30%wt. The ionic liquid can be easily removed by washing mixture fibers with 60 °C hot water when fibers exit from the twin-screw extruder die.

### 4.5. Rheological Analysis of PAN and Lignin/PAN Blends

Rheological characterization of all melt-spun samples was conducted using Discovery Hybrid Rheometer (TA Instruments, New Castle, DE, USA) in parallel plate geometry, with a 25-mm plate on top and Peltier plate on the bottom with a 500-μm gap size in between. In one set of experiments, isochronal temperature scans of polymer melts were performed at 1 rad/s. The Peltier plate was maintained at 150 °C before loading the samples. A thin layer of silicone oil was applied to cover the exposed surface of polymer melt to prevent moisture absorption after the polymer samples were loaded between the parallel plates. Ten minutes after loading the samples (the initial temperature at 150 °C), samples were heated to 200 °C, followed by cooling down from 200 °C to 150 °C. The heating and cooling rates were controlled at 2 °C/min, and the rheological data were continuously collected. In the second set of experiments, dynamic frequency sweep measurements from 0.1 to 100 rad/s were conducted at a temperature range of 150 °C to 200 °C, with an increment of 10 °C. The temperature range was chosen to provide an understanding of polymer melts during expected fiber processing window, which is below the initial degradation temperature of PAN polymer at around 220 °C. In the third set of experiments, steady-state shear rheology was determined under a shear rate from 0.01 s^−1^ to 10 s^−1^ at a temperature of 180 °C. This was done to demonstrate the equivalence between the steady and the dynamic shear viscosities.

### 4.6. Thermal Analysis of PAN and Lignin/PAN Blends

The thermal melting behavior of PAN and PAN-lignin blends in [BMIM]Cl was evaluated using a differential scanning calorimeter (DSC; DSC 214 Polyma, Netzsch, Germany). Approximately 5–10 mg of the pure or blended samples were heated from 30 to 220 °C at a heating rate of 10 °C/min under a nitrogen atmosphere with a purge flow of 50 mL/min. The DSC parameters of the endotherms, including initiation temperature, final temperature, and peak temperature were obtained and analyzed using Proteus software (v7, Netzsch, Germany).

## 5. Conclusions

These studies have shown significant evidence for polymeric entanglements and intermolecular interactions with an increase in the concentration of butyrated lignin in PAN at melt spinning temperatures. It was also found that, while an increase in BL resulted in increased yield stress at different temperatures, the optimal melt spinning temperatures varied with varying BL concentration primarily to maintain good thermo-reversibility and pseudo-plastic behavior during heating and cooling zones. Furthermore, melt spinning PAN-BL blends above the predicted gel points can result in significant decreases in the homogeneity of the fibers and most likely, mechanical properties of any carbon fiber product. The presence of BL also had an effect on the cyclization reactions between PAN and [BMIM]Cl, resulting in a deviation of linearity in the solution activation energy model that has been previously observed for PAN in [BMIM]Cl. Further studies on the chemical interactions between the modified (butyrated) biorefinery lignin with the PAN-[BMIM]Cl solution using techniques such as FTIR [13] and NMR [27] will most likely improve our understanding of the polymer blends and its effect on rheology. Finally, the research discussed in this manuscript provides important data of relevance for the future development in melt spinning of PAN precursors with biomass components such as lignin to produce possible carbon fiber precursors. Optimization of the melt-spinning conditions could significantly reduce the solvent requirements and improve the overall process economics.

## Figures and Tables

**Figure 1 molecules-24-02650-f001:**
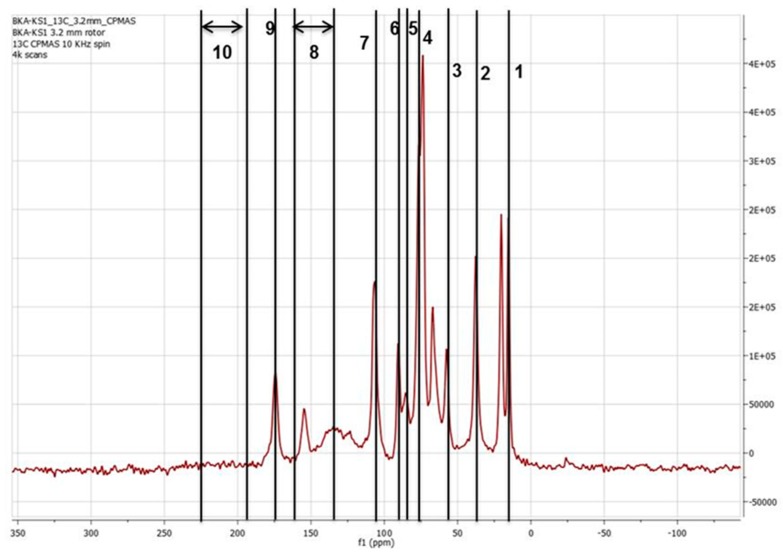
^13^C CP/MAS NMR spectra of butyrated biorefinery lignin (BL). Description of components/linkages are as follows: 1 → 21–24 ppm (CH_3_ in acetyl groups in hemicellulose); 2 →30–40 ppm (alkyl CH & CH_2_ in lignin subunits); 3 → 56 ppm (methoxy groups in lignin G & S units); 4 → 82 ppm (C_6_ carbon atoms in carbohydrates); 5 → 89 ppm (C_4_ carbon atoms in carbohydrates); 6 → 105 ppm (C_1_ carbon atoms in carbohydrates); 7 → 143–167 ppm (unsubstituted olefinic or aromatic C-atoms with OH or RO substituents); 8 → 169–195 ppm (esters & carboxylic acids); 9 → 173 ppm (CH in acetyl groups in hemicellulose); and 10 → 195–225 ppm (carbonyl groups in lignin).

**Figure 2 molecules-24-02650-f002:**
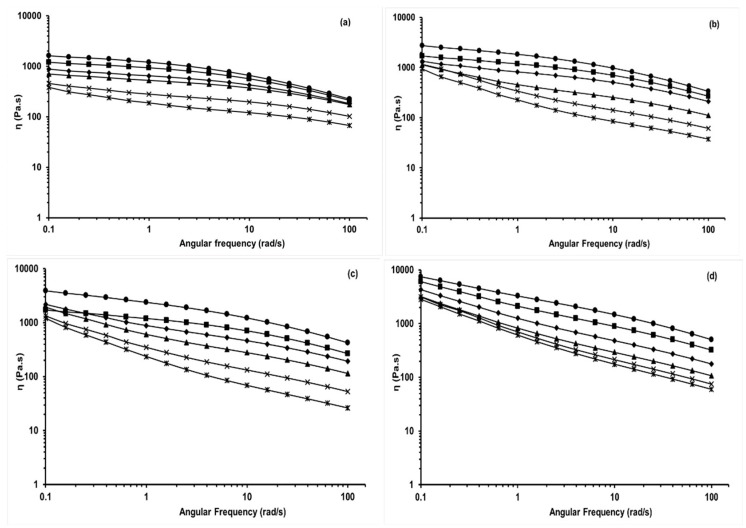
Complex viscosity (η) as a function of angular frequency (rad/s) at different temperatures (150–200 °C) for (**a**) polyacrylonitrile (PAN); (**b**) PAN-10%wt BL; (**c**) PAN-20%wt BL; and (**d**) PAN-30%wt BL; ●—150 °C; ■—160 °C; ◆—170 °C; ▲—180 °C; x—190 °C; ӿ—200 °C.

**Figure 3 molecules-24-02650-f003:**
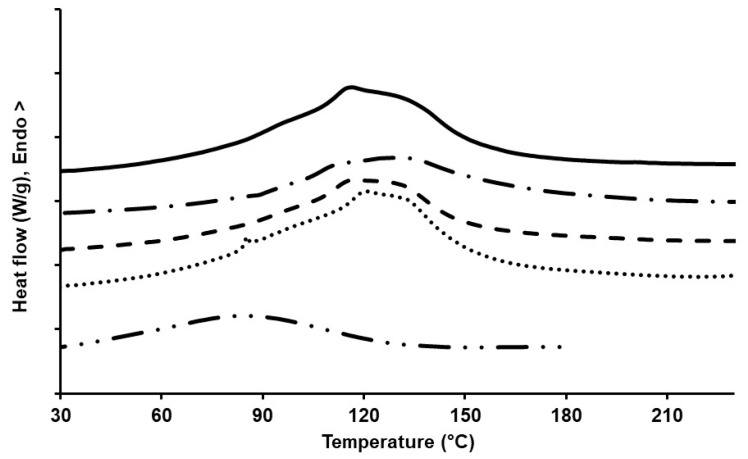
DSC curves of PAN/BL blends with different concentrations of BL and a reference curve for BL; ▬ PAN; ▬ ▪ PAN-10%wt BL; ▬ ▬ PAN-20%wt BL; ▪▪▪ PAN-30%wt BL; ▬▪▪ BL.

**Figure 4 molecules-24-02650-f004:**
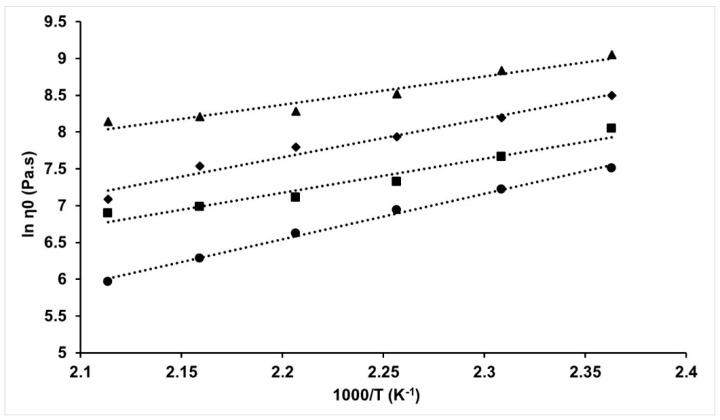
Arrhenius plot for Zero-shear viscosity of PAN and PAN-BL blends as a function of temperature; ●—PAN; ■—PAN-10%wt BL; ◆—PAN-20%wt BL; ▲—PAN-30%wt BL.

**Figure 5 molecules-24-02650-f005:**
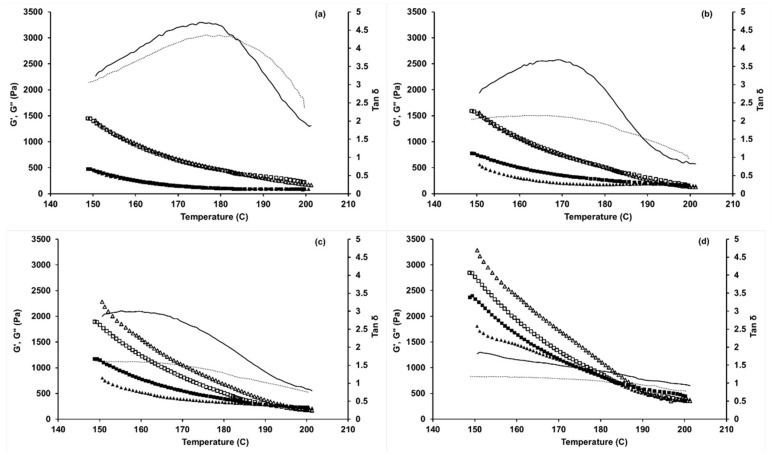
Storage modulus (G’), loss modulus (G′′) and loss tangent (tan δ) of (**a**) PAN; (**b**) PAN-10%wt BL; (**c**) PAN-20%wt BL; and (**d**) PAN-30%wt BL blends as a function of temperature between 150 and 200 °C. ▲-G’ (heating); ■-G’ (cooling); Δ-G′′ (heating); □-G′′ (cooling); ▬ tan δ (heating); ▪▪▪ tan δ (cooling).

**Figure 6 molecules-24-02650-f006:**
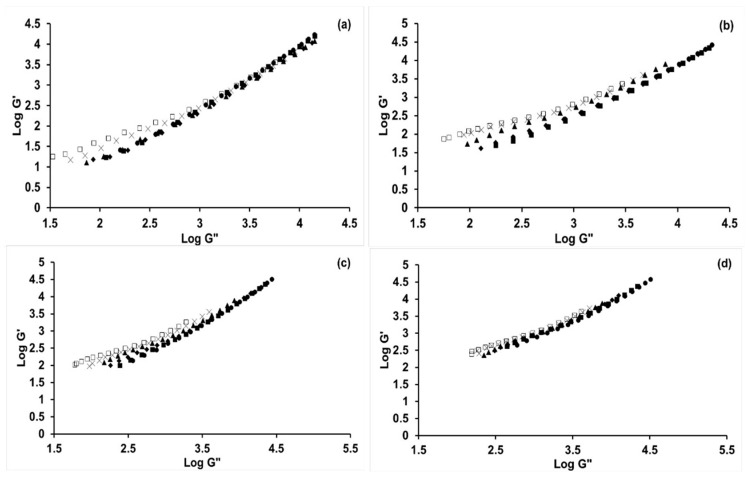
Log-log G’-G′′ plot (or Han plot) as a function of temperature for (**a**) PAN; (**b**) PAN-10%wt BL; (**c**) 20%wt BL; and (**d**) 30%wt BL in [BMIM]Cl; ●—150 °C; ■—160 °C; ◆—170 °C; ▲—180 °C; X—190 °C; □—200 °C.

**Figure 7 molecules-24-02650-f007:**
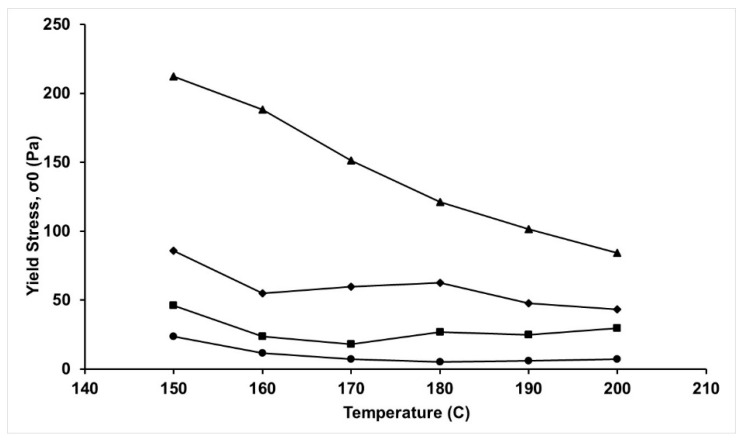
Solution yield stress for PAN and PAN-BL blends as a function of temperature; ●—PAN; ■—PAN-10%wt BL; ◆—PAN-20%wt BL; ▲—PAN-30%wt BL.

**Table 1 molecules-24-02650-t001:** Characteristics of Butyrated Biorefinery Lignin Stream.

Compositional Analysis ^a^				M_n_ (kDa) ^b^	M_w_ (kDa) ^b^	PDI ^b^	T_g_ (°C) ^c^	C_p_ (J/g) ^c^
Glucan (%wt)		Lignin (%wt)						
30.3		65.8		127.5	284.5	2.23	103.2	97.13
%C	%H		%N					
50.98	6.03		11.23					

^a^ Compositional analysis of biorefinery lignin was done before butyration. ^b^ M_n_ and M_w_ refer to the number-average and weight-average molecular weight of the lignin; PDI refers to polydispersity—data obtained from size exclusion chromatography. ^c^ T_g_ refers to glass transition temperature, and C_p_ refers to the heat capacity of the lignin measured using differential scanning calorimetry.

**Table 2 molecules-24-02650-t002:** The slope of the Han plot at different temperatures and BL blending with PAN in [BMIM]Cl.

Polymer Solution	Slope of Han Plot (Dimensionless)					
	150 °C	160 °C	170 °C	180 °C	190 °C	200 °C
PAN	1.47	1.45	1.36	1.32	1.16	1.01
PAN-10% BL	1.34	1.31	1.21	1.09	0.90	0.82
PAN-20% BL	1.27	1.20	1.08	0.98	0.91	0.76
PAN-30% BL	1.10	1.00	0.95	0.92	0.83	0.80

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
