# Peer review of "Rheology of Polyacrylonitrile/Lignin Blends in Ionic Liquids under Melt Spinning Conditions"

_molecules, 2019, doi:10.3390/molecules24142650_

Round 1

Reviewer 1 Report

Page 1 – Line 19 -21

Authors cannot justify the limited success of technical ligninon this application based on harmful chemicals. There are extraction processes, which do not use harmful chemical during extraction. With all respect, this sentence sound more “marketing” than “science”. The authors should rewrite the sentence, or describe in details how the “harmful chemical” affect the success of this kind of application.

Page 2 – Line 61.

Authors claim that :

The complex heterogeneous structure of lignin with randomly cross-linked phenylpropane units into 3-D network makes it difficult to spin and spool into fibers without modification.

It is not clear, how the (WEx) lignin could overcome these advantages, since it has extremely high molar mass (Table 1)? Why the authors did not applied the same methology (ionic liquids) with common technical lignins (from kraft, organosolv and sulphite processes)?

Page 3 – Line 120.

Once again, it sounds strange. There are some technical grades of lignin in the industrial market, which are largely available, with negligible cellulose content! Without the need for removing cellulose!!

Page 4 – Line 124

Such separation and purification processes can result in an increase in the cost of lignin, which can remove the benefits of a low-cost feedstock for carbon fiber production.

Not really! Unmodified technical kraft lignin (with low cellulose content) is available at USD 1.00/kg or less.

In addition, in this study, the butyration mehod is applied, which can elevate the lignin cost.

 Page 5 – Line 187

If I correct understood, the DSC curves are related to a blend of PAN/[BMIM]Cl with further lignin addition. In this case, blend of neat PAN/lignin must be provided and analysed by DSC, in order to verify the [BMIM]Cl influence?

Why a initial thermal cycle was not applied, in order to eliminate the thermal history o the samples?

How the authors obtained the lignin Tg from the Figure 3? I cannot see any transition, which can be related to Tg at 103 °C.

Author Response

We thank the reviewer for the comments and suggestions and for the opportunity to discuss the merits of the results obtained from this study. From the comments, we understand that there were some misunderstanding about the scope of this manuscript and we have made an effort to answer the questions and keep the revisions within the scope of the study discussed in the manuscript.

Page 1 – Line 19 -21

Authors cannot justify the limited success of technical lignin on this application based on harmful chemicals. There are extraction processes, which do not use harmful chemical during extraction. With all respect, this sentence sound more “marketing” than “science”. The authors should rewrite the sentence, or describe in details how the “harmful chemical” affect the success of this kind of application.

We thank the reviewer for his comment but there is published research where the lignin structure has shown to significantly affect its use in carbon fiber production. We have given some literature within the manuscript that outline it but we understand why the statement as it currently stands might look like marketing versus science. So, we have given another reference that explicitly discusses this. In short, regular biomass pretreatment processes use bases or acids and research has shown that in presence of base, the lignin undergoes repolymerization and recondensation affecting its ability to be homogeneously blended into carbon fibers. In case of using sulfuric acids or sulfite pulping, the lignosulfonates have been shown to be weaker due to the presence of sulfur. These acids and bases are corrosive in nature and requires specialized treatment of the wastewater from the biorefineries and hence, listed as harmful. However, the goal of this study is not to compare between different biomass pretreatment processes and their effect on lignin but to use the lignin that we obtained from a specific chemical-free pretreatment method (structure and characteristic previously established in Rana et al., 2015) and to study its rheology when blended with PAN under melt spinning conditions.

Page 2 – Line 61.

Authors claim that :

The complex heterogeneous structure of lignin with randomly cross-linked phenylpropane units into 3-D network makes it difficult to spin and spool into fibers without modification.

It is not clear, how the (WEx) lignin could overcome these advantages, since it has extremely high molar mass (Table 1)? Why the authors did not applied the same methology (ionic liquids) with common technical lignins (from kraft, organosolv and sulphite processes)?

We appreciate the reviewer’s comment since we understand why the statement can be confusing. This sentence is aimed at indicating why 100% lignin used directly for producing carbon fibers (without PAN) had failed and this has been significantly shown by several researchers (cited earlier in manuscript) as well as shown by extensive 5-yr research done by US Department of Energy. This paragraph aimed to explain why researchers are currently looking to instead blend lignin with PAN to reduce the cost instead.

With respect to using ionic liquids for lignin extraction, the goal of this manuscript is to look at rheology of PAN-biorefinery lignin blends under melt spinning conditions (many researchers have traditionally used solution spinning for producing carbon fibers and it is textbook knowledge that using melt spinning instead of solution spinning can both reduce significant solvent consumption during carbon fiber production as well as improve interactions between lignin and PAN in the blends. However, data is limited in this field and the study addresses it). The goal of the manuscript is to use biorefinery lignin obtained from a pretreatment process (that has previously shown to have very similar structure as native lignin) and test its ability to be blended with PAN to produce carbon fibers with minimal treatment. (butyration will be explained below).

Page 3 – Line 120.

Once again, it sounds strange. There are some technical grades of lignin in the industrial market, which are largely available, with negligible cellulose content! Without the need for removing cellulose!!

We agree with the reviewer that there is several technical grades of lignin in the industrial market that do not have cellulose but as indicated in the “lignin characterization” sub-section of the Results section, the lignin structure after current chemical-free pretreatment is significantly methoxylated making it a good lignin substrate for carbon fiber production. Whether the current lignin is better than technical lignin does not fall within the scope of this study and is a separate study in itself and was never cited as the main focus of the manuscript. Our discussion primarily aims to describe the process and mainly, the type and structure of lignin that was obtained to perform this study. There are no previous studies that have shown any disadvantage of having cellulose in the lignin and we did not see a significant impact ourselves, especially due to the high melt spinning temperatures. The use of this biorefinery lignin blended with PAN has shown some very interesting rheological properties that qualify its use in carbon fiber production, as outlined throughout the manuscript.

Page 4 – Line 124

Such separation and purification processes can result in an increase in the cost of lignin, which can remove the benefits of a low-cost feedstock for carbon fiber production.

Not really! Unmodified technical kraft lignin (with low cellulose content) is available at USD 1.00/kg or less.

In addition, in this study, the butyration mehod is applied, which can elevate the lignin cost.

We agree with the reviewer on the cost of technical kraft lignin but as cited earlier in the literature, there is abundant literature that has cited the disadvantage of using kraft lignin for carbon fiber production. However, that being said, it is common knowledge that separation and purification processes in a biorefinery can account to 20-50% of the total processing costs. The use of biorefinery lignin (without the complete separation from cellulose) did not show any significant disadvantage related to its blending with PAN and in fact, the rheological data was found to be better (discussed in section on Storage-loss modulus and Han plots) up to 20wt% blending. Again, as indicated above, the goal of this study is to understand the rheological characteristics of such a biorefinery lignin applied for carbon fiber production and not to optimize the type of lignin used. However, we agree with the reviewer on the confusion caused by this statement unrelated within the scope of the study and hence, we replaced it in the manuscript.

With respect to butyration method used, due to the high temperatures used in melt spinning, it is necessary to esterify the lignin (irrespective of type of lignin used) and this standard process has been summarized earlier in the Introduction section (literatures Ding et al., 2016; Thunga et al., 2014). There is a general confusion when lignin-based carbon fibers are discussed since most existing research is done using solution spinning. However, the manuscript focuses on blending PAN and lignin under melt spinning conditions (as indicated in title and the introduction section) and lignin esterification is a part of this process.

 Page 5 – Line 187

If I correct understood, the DSC curves are related to a blend of PAN/[BMIM]Cl with further lignin addition. In this case, blend of neat PAN/lignin must be provided and analysed by DSC, in order to verify the [BMIM]Cl influence?

Why a initial thermal cycle was not applied, in order to eliminate the thermal history o the samples?

How the authors obtained the lignin Tg from the Figure 3? I cannot see any transition, which can be related to Tg at 103 °C.

We thank the reviewer for the comment but we have not indicated anywhere in the manuscript that pure lignin Tg was obtained from Figure 3. Instead, Figure 3 shows the DSC curve for butyrated lignin and neat PAN and their blends. Considering that the study was based on blending PAN with butyrated lignin, we did not feel the use of adding separate DSC curve for pure lignin. Preliminary experiments with an initial thermal cycle did not show any difference in the DSC curves and after repeating experiments, we determined that it did not have any impact on the trends and data discussed in the manuscript.

Reviewer 2 Report

I have made a large number of comments in the annotated pdf file, most of which are minor English corrections. In some places I have asked for clarification.

Author Response

We thank the reviewer for the comments, suggestions and grammatical corrections at different parts of the manuscript. We have addressed every suggestion made by the reviewer and would like to address few questions/comments indicated by the reviewer in the manuscript.

Confirm the PAN/[BMIM]Cl blending ratio in the manuscript.

As indicated, we kept the ratio of the polymer (or polymer blend) in the ionic liquid consistent throughout the study at 50wt%. We confirmed it in the experimental section of the manuscript.

Improving clarity of Figure 5.

We have high resolution graphs but when constrained within the small size, this was the best clarity that we could get. However, based on the reviewer’s suggestion, we would like to work with the journal to get the best clarity of the graphs in final manuscript.

Comment on the decreasing trend in activation energy of the blends

We had an increased discussion on the trends in the “discussion” section of the manuscript but based on the reviewer’s suggestions, we have indicated our hypothesis and where it is discussed in the manuscript.

Clarify “thermo-reversibility”

We have clarified “thermo-reversibility” as suggested by the reviewer.

Round 2

Reviewer 1 Report

My suggestion it to accept in the present form.